

# A simple equation to estimate body fat percentage in children with overweightness or obesity: a retrospective study

Ernesto Cortés-Castell[1], Mercedes Juste[1], Antonio Palazón-Bru[2], Laura Monge[1], Francisco Sánchez-Ferrer[1] and María Mercedes Rizo-Baeza[3]

[1] Pharmacology, Pediatrics and Organic Chemistry Department, Miguel Hernández University, San Juan de Alicante, Alicante, Spain
[2] Clinical Medicine Department, Miguel Hernández University, San Juan de Alicante, Alicante, Spain
[3] Nursing Department, University of Alicante, San Vicente del Raspeig, Alicate, Spain

## ABSTRACT

**Background**. Dual-energy X-ray absorptiometry (DXA) provides separate measurements of fat mass, fat-free mass and bone mass, and is a quick, accurate, and safe technique, yet one that is not readily available in routine clinical practice. Consequently, we aimed to develop statistical formulas to predict fat mass (%) and fat mass index (FMI) with simple parameters (age, sex, weight and height).

**Methods.** We conducted a retrospective observational cross-sectional study in 416 overweight or obese patients aged 4–18 years that involved assessing adiposity by DXA (fat mass percentage and FMI), body mass index (BMI), sex and age. We randomly divided the sample into two parts (construction and validation). In the construction sample, we developed formulas to predict fat mass and FMI using linear multiple regression models. The formulas were validated in the other sample, calculating the intraclass correlation coefficient via bootstrapping.

**Results**. The fat mass percentage formula had a coefficient of determination of 0.65. This value was 0.86 for FMI. In the validation, the constructed formulas had an intraclass correlation coefficient of 0.77 for fat mass percentage and 0.92 for FMI.

**Conclusions**. Our predictive formulas accurately predicted fat mass and FMI with simple parameters (BMI, sex and age) in children with overweight and obesity. The proposed methodology could be applied in other fields. Further studies are needed to externally validate these formulas.

## INTRODUCTION

During childhood and adolescence, there is a balanced growth of the different body components: skeletal muscle, fat, bone and viscera. Obesity involves an increase in body weight combined with an imbalance between these components, with a higher proportion of body fat (*Ballabriga & Carrascosa, 2006*), and is defined as an excess of body fat relative to total body mass (*Himes & Dietz, 1994*). There is nearly universal consensus in defining overweight as a body mass index (BMI ($kg/m^2$)) between the 85th and 95th percentiles and obesity as a BMI at the 95th percentile or greater (*Power, Lake & Cole, 1997*; *Prentice,*

Corresponding author
Antonio Palazón-Bru,
antonio.pb23@gmail.com

*1998*; *Serra Majem et al., 2002*; *Cole & Lobstein, 2012*). Determinants of body fat in children and adolescents include socioeconomic factors, education of the mother, physical activity and physical fitness (*Moliner-Urdiales et al., 2009*; *Van Sluijs et al., 2010*; *Gómez-Martínez et al., 2012*).

Age- and sex-specific BMI percentile values are easy to calculate and is strongly correlated with the body fat percentage, especially for high BMI values (*Colomer, 2004*; *Koplan, Liverman & Kraak, 2005*; *Krebs et al., 2007*). But in clinical practice, it may be more useful to combine age- and sex-specific BMI values with a body fat assessment capable of detecting a high degree of adiposity to avoid classifying individuals whose high BMI is attributable to a relatively greater fat-free mass or an athletic build as obese (*Serra Majem et al., 2002*; *Whitlock et al., 2005*). Since BMI does not follow a normal distribution in the pediatric age group, several indices have been proposed as alternatives. Of these, the recently described inverted BMI (iBMI ($cm^2$/kg)) has been referred to as the most useful and accurate proxy for body fat in adults (*Nevill et al., 2011*). A recent study in the pediatric age group that used the fat mass determined by dual-energy X-ray absorptiometry (DXA) as a reference found that the iBMI followed a normal distribution and was a good predictor of body fat, as was BMI, with iBMI accounting for a greater amount of the variance (*Duncan et al., 2014*). Although various equations have been proposed to calculate body fat based on skinfolds they are not recommended for use in the pediatric population (*Almeida et al., 2016*; *Truesdale et al., 2016*).

DXA enables the independent assessment of fat mass, boneless fat-free mass and bone mass (*Fields & Goran, 2000*; *Kehayias & Valtueña, 2001*) and their distribution in each region of the body. DXA is considered a fast, accurate, and safe method for assessment of body fat (*Ellis, 2001*), as it is free of the drawbacks of computerized axial tomography and magnetic resonance imaging (*Goulding et al., 1996*). However, since it is not widely available, this technique is not suitable for large-scale and longitudinal studies. For this reason, several indices (BMI, iBMI, conicity index, etc.) have been proposed to estimate the value obtained by DXA. Other authors have constructed multivariable prediction models that estimate body fat from different skinfolds. These methods have been validated by contrasting the mathematical formula with DXA (reference standard) (*Silva et al., 2013*; *Jensen, Camargo & Bergamaschi, 2016*). Nonetheless, they present great difficulty in routine clinical practice because skinfolds are not as simple to measure or as reproducible as the weight and height of a child. For all these reasons, we aimed to develop and internally validate (intraclass correlation coefficient (ICC) and bootstrapping) a statistical model based on simple parameters (BMI, iBMI, sex and age) to predict body fat. Our goal was a simple tool that could be applied in clinical practice to assess adiposity in children and adolescents.

## MATERIALS & METHODS

### Study population

Patients referred for nutritional problems to the Nutrition, Growth and Metabolism Unit of the Department of Pediatrics of San Juan de Alicante University Hospital. This hospital

covers an area of about 220,000 inhabitants in the province of Alicante, which is located in the southeast of Spain and has a total of 1,843,589 inhabitants (*Instituto Nacional de Estadítica, 2016*). The health system is universal and free for both children and adults. The prevalence of childhood obesity in the province of Alicante is approximately 13.5–18.8% (*Ruiz & Pérez et al., 2008*). The criterion for referral by the primary care pediatrician was overweight/obesity.

## Study design and participants

We conducted a cross-sectional observational study in patients aged 4–18 years who met the following inclusion criteria: having had an initial referred visit to the Nutrition, Growth and Metabolism Unit of San Juan de Alicante University Hospital and a DXA assessment prior to the implementation of dietary measures and lifestyle changes. The exclusion criteria encompassed excess weight secondary to causes other then high caloric nutrition, such as growth hormone deficiency with hormone replacement therapy, syndromic obesity, obesity secondary to other diseases such as hypothyroidism, protracted treatment with corticosteroids or other drugs that could influence energy intake or expenditure, precocious puberty, or neurologic or neuromuscular disorders preventing the patient from walking or exercising normally. The data were collected between July 2007 and July 2016, and all the study children were Caucasian.

## Variables and measurements

Our main outcome variables were body fat (% total body fat) and fat mass index (FMI, in $kg/m^2$). These were measured with a General Electric Lunar DPXN $PRO^{TM}$ DXA densitometer (GE Healthcare, Little Chalfont, Buckinghamshire, United Kingdom). The software of this device is able to obtain measurements for the total weight and percentage of body fat. These parameters were used as the reference standard measurements.

As secondary variables we used BMI (in $kg/m^2$), sex (male or female) and age (in years). To obtain the anthropometric measurements (weight and height), we followed a standardized protocol with a stadiometer accurate to 0.1 cm and a SECA scale accurate to 0.1 kg. Each value was measured twice by a single individual, and the mean of the two measurements was used in the analysis.

## Sample size calculation

The sample collected during the study period was 416 children. This sample was randomly divided into two equal parts ($n = 208$). The first group was used to construct a predictive model (multivariable linear regression) and the second group was used to validate it. Construction: To construct a multiple linear regression model we must look at the relationship between the number of subjects and the number of predictors (subject-to-variable ratio). As a heuristic rule, it was considered that this ratio should be at least 50, which allowed the introduction of four explanatory variables in the predictive model. Validation: The sample consisted of the other 208 subjects. This sample size was very satisfactory in obtaining excellent predictions, since the constructed models had four predictors and coefficients of determination of 0.65 and 0.86 for fat percentage and FMI

respectively (*Knofczynski & Mundfrom, 2008*). The coefficient of determination values were obtained in the construction sample and were used to determine the sample size of the validation.

## Statistical analysis

Continuous variables (BMI, FMI, age and body fat) were summarized using means and standard deviations. To describe the sex variable, we calculated absolute and relative frequencies. To compare the homogeneity of the construction and validation samples, we performed the $t$-test and the Pearson's chi-squared test. We calculated the following exponentiations: bases (BMI and age) and exponents ($-2$, $-1$, 1 and 2). All the interactions between sex and the exponentiation variables were obtained. In the construction sample ($n = 208$) we aimed to construct a linear regression model to predict body fat. Taking into account that we could only introduce 4 explanatory variables (subject-to-variable ratio: one per each fifty subjects) and had 17, we applied the following algorithm in order to select them: we obtained all the possible combinations of 1, 2, 3 and 4 elements from the total (17 variables): 3,213 combinations. In other words, we analyzed 6426 different models ($3,213 \times 2 \approx 6,500$). In each combination we estimated the linear regression model with them in order to predict the body fat and calculated its coefficient of determination. The combination/model with the maximum coefficient was then selected. The goodness-of-fit of the model was assessed with the ANOVA test. The model was internally validated using bootstrap methodology in the validation sample, calculating the intraclass correlation distribution (two fixed judges (the predicted and observed value) with absolute agreement in the ratings). In addition, the scatter plot with the estimation and the real parameter was obtained. Finally, we also calculated the Bland & Altman limits of agreement (*Bland & Altman, 1986*). The same process was applied for FMI. We set the level of statistical significance at 0.05. The statistical software used was IBM SPSS Statistics 19 and R 2.13.2.

## Ethical considerations

This study adhered to the ethical principles of the Declaration of Helsinki, and only involved the performance of procedures used in everyday clinical practice. The data were processed safeguarding anonymity and confidentiality, and the study was approved by the corresponding Ethics Committee (Comité Ético de Investigación Clínica del Hospital Universitario de San Juan de Alicante, ref 16/305). Informed consent was not requested from the parents for this study, since it was part of routine clinical practice without any type of intervention. The Ethics Committee approved this procedure.

## RESULTS

We analyzed a total sample of 416 children divided into two parts: 208 children in each group (construction and validation). Table 1 shows the descriptive analysis obtained for each group. Of note were a mean body fat of 43% and 11.7 kg/m$^2$ for FMI. No differences were observed in the groups, as all the $p$-values were greater than 0.05.

The optimal model for body fat had a coefficient of determination of 0.65 and the following formulas for the body fat estimation (Table 2):

**Table 1 Descriptive and comparative analysis for construction and validation samples.**

| Variable | Construction sample $n = 208$ $n(\%)/x \pm s$ | Validation sample $n = 208$ $n(\%)/x \pm s$ | p-value[b] |
|---|---|---|---|
| Body fat (%)[a] | $43.2 \pm 8.2$ | $43.1 \pm 7.9$ | 0.914 |
| FMI (kg/m$^2$)[a] | $11.7 \pm 3.5$ | $11.7 \pm 3.5$ | 0.986 |
| BMI (kg/m$^2$) | $26.6 \pm 4.0$ | $26.7 \pm 4.3$ | 0.918 |
| Male sex | 111(53.4) | 113(54.3) | 0.844 |
| Age (years) | $11.4 \pm 2.8$ | $11.3 \pm 2.8$ | 0.777 |

Notes.

Abbreviations: BMI, body mass index; FMI, fat mass index; $n(\%)$, absolute frequency (relative frequency); x ± s, mean ± standard deviation.

[a] measured by dual -energy X-ray absorptiometry.

[b] Pearson's chi-squared test (qualitative variables) or $t$-test (quantitative variables).

**Table 2 Optimal multivariate models in order to predict our main outcome variableses,**

| Variable | Body fat model B (95% CI) | p-value | FMI model B (95% CI) | p-value |
|---|---|---|---|---|
| Constant | 62.627 (59.828,65.426) | <0.001 | 18.655 (14.106,23.203) | <0.001 |
| BMI$^{-2}$ | $-11245.580(-12997.421, -9493.738)$ | <0.001 | N/M | N/M |
| BMI$^{-1}$. Male sex | $-259.114(-443.278, -74.949)$ 0.006 | N/M | N/M | |
| Age.Male sex | 2.310 (0.887,3.732) | 0.002 | 0.112 $(-0.010, 0.234)$ | 0.073 |
| Age$^2$. Male sex | $-0.151 (-0.220, -0.082)$ | <0.001 | $-0.018 (-0.027, -0.009)$ | <0.001 |
| BMI$^2$ | N/M | N/M | 0.007 (0.005,0.009) | <0.001 |
| BMI$^{-1}$ | N/M | N/M | $-293.601 (-374.123, -213.079)$ | <0.001 |

Notes.

Abbreviations: B, regression coefficient; BMI, body mass index; CI, confidence interval; FMI, fat mass index; N/M, not in the model.

Goodness-of-fit of the models (ANOVA test): (1) body fat: $F = 94.404, p < 0.001$; (2) FMI: $F = 319.299, p < 0.001$.

(A) Boys: $62.627 - 11245.580.\text{BMI}^{-2} - 259.114.\text{BMI}^{-1} + 2.310.\text{Age} - 0/151.\text{Age}^2$.

(B) Girls: $62.627 - 11245.580\text{BMI}^{-2}$.

For FMI the optimal model had a coefficient of determination of 0.86 and the formulas for the estimation of this parameter were (Table 2):

(A) Boys: $18.655 + 0.007.\text{BMI}^2 - 293.601.\text{BMI}^{-1} + 0.112.\text{Age} - 0.018.\text{Age}^2$.

(B) Girls: $18.655 + 0.007.\text{MI}^2 - 293.601.\text{BMI}^{-1}$.

The intraclass correlation distribution is shown in Fig. 1 (internal validation). Regarding the adjustment between estimations and real values, Fig. 2 illustrates that both values were very similar. Finally, the Bland & Altman procedure has a good level of agreement, because most of the points were between the limits of agreement (dashed lines) (Fig. 3). So that our equations may be easily applied, they have been included in Table S1.

# DISCUSSION

## Summary

We constructed and validated two predictive models to determine body fat and FMI in children with overweight and obesity, using the values obtained by DXA as the reference standard. Both models showed excellent accuracy (ICC > 0.75) (*Cicchetti, 1994*).

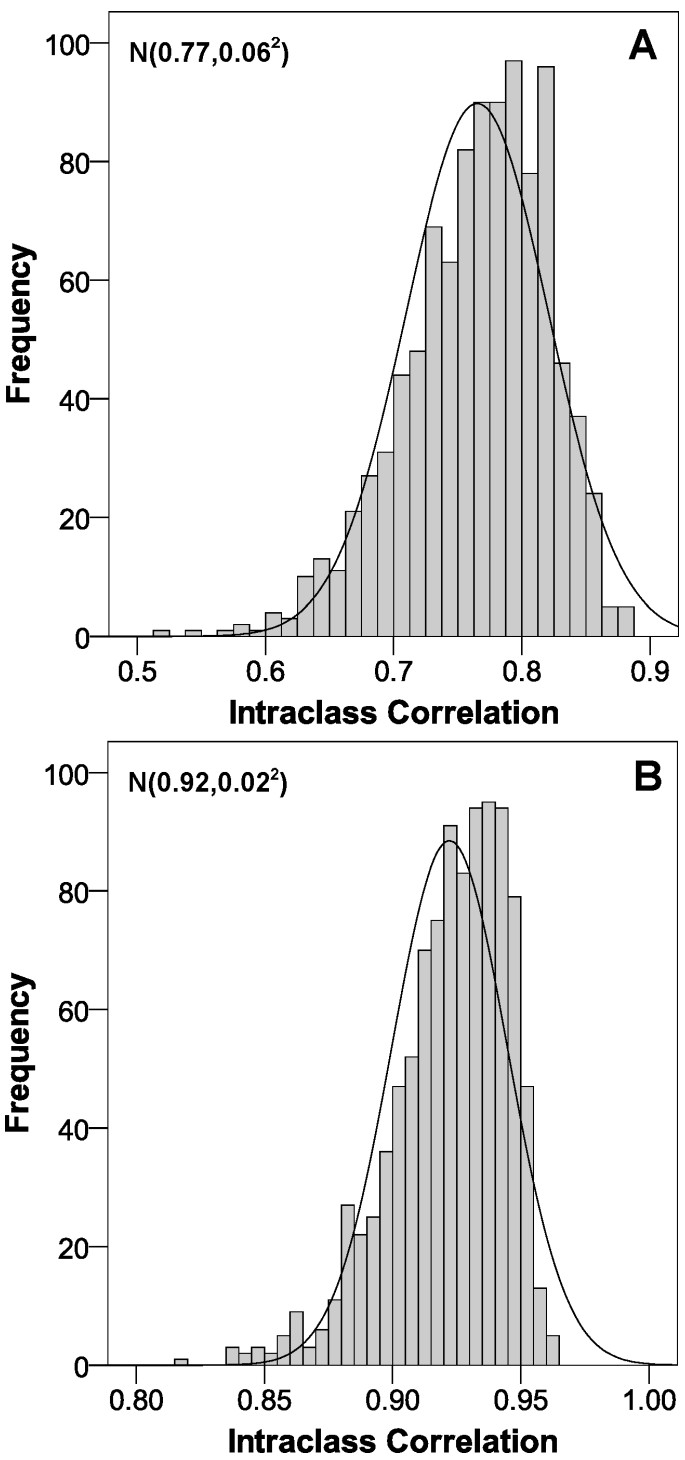

**Figure 1** **Intraclass correlation coefficient distribution of the estimated parameters obtained through the bootstrap methodology.** (A) body fat; (B) fat mass index.

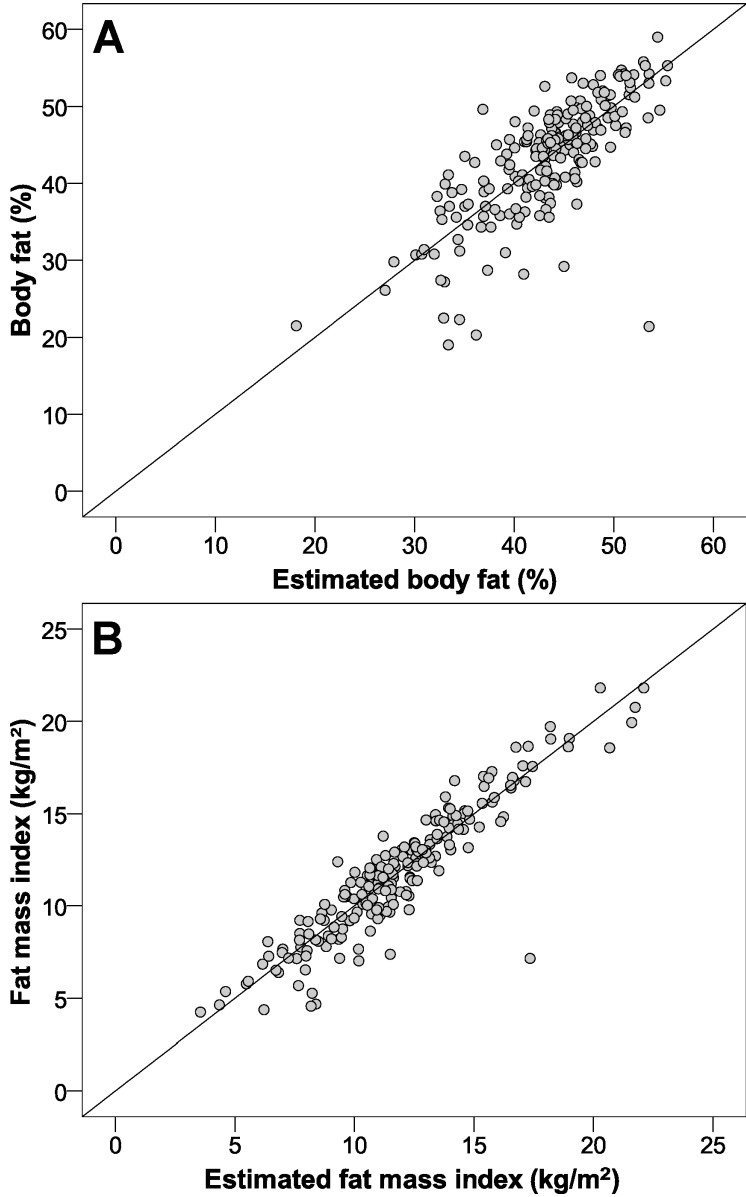

**Figure 2 Scatter plot to show the adjustment between the proposed formulas (estimations) and the real parameters in the validation sample.** (A) body fat; (B) fat mass index.

## Strengths and limitations of the study

The main strength of the study is the easy and cost-effective calculation of body fat percentage and FMI based on the widely used standard anthropometric measurements (height and weight), age and sex. Although more complex techniques, such as DXA, consume little time and few resources and involve a very low radiation exposure, they are not widely available in health care settings and are more complicated and costly to implement. We also underscore the statistical methodology followed for its construction, which took into account the subject-to-variable ratio of about 6,500 models. For validation, the sample size was adequate to obtain excellent predictions.

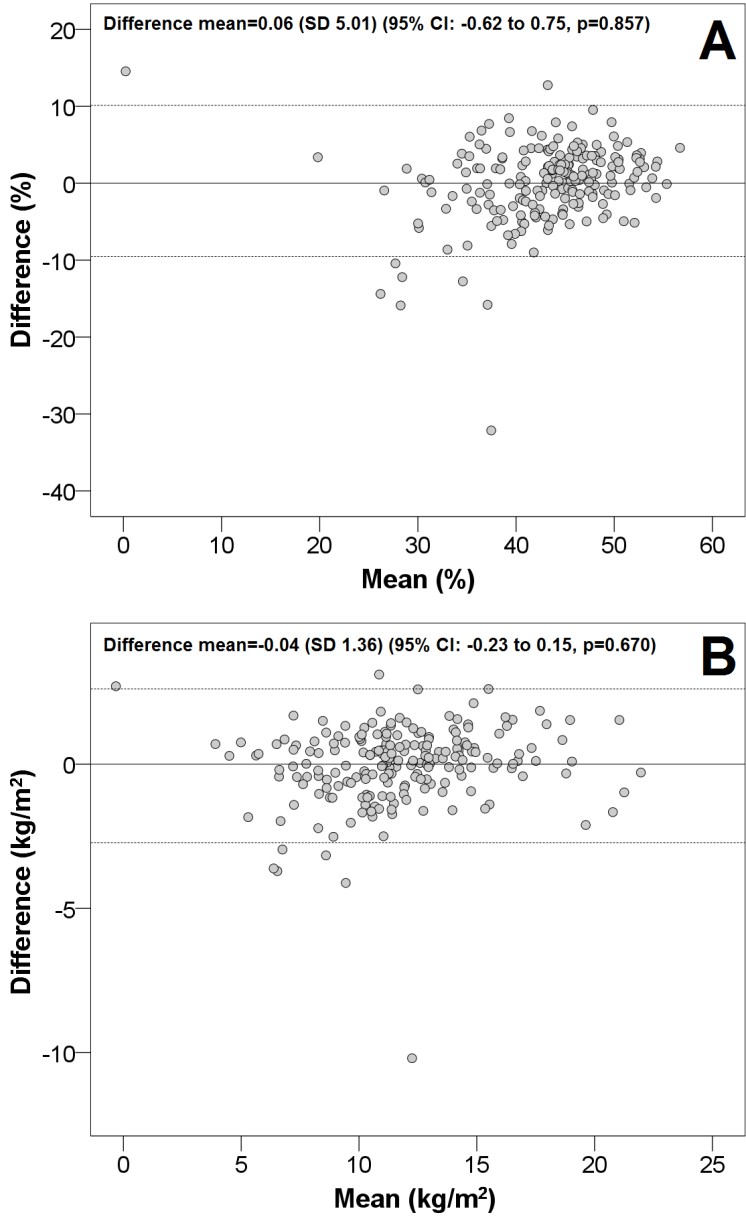

**Figure 3** **The Bland & Altman plot for our proposed formulas.** CI, confidence interval; SD, standard deviation. The dashed lines are the limits of agreement (±1.96 SD). (A) body fat; (B), fat mass index.

Regarding limitations, the study was clearly lacking in normal-weight cases (selection bias), but due to ethical and cost-of-care considerations, body fat measurement using DXA was not justified in children of normal weightand was only performed for other clinical purposes, such as in patients that required a bone mineral density assessment. On the other hand, information bias was minimized through the use of fully validated and calibrated devices, and confounding bias was minimized through the estimation of multivariable models. Thus, our models can only be applied in children with overweight or obesity. Furthermore, although DXA is not considered the gold standard for the assessment

of body fat because a four-component model is better (*Sopher et al., 2004*; *Sopher, Shen & Peitrobeilli, 2005*; *Williams et al., 2006*), DXA is generally considered to provide the most valid estimates in clinical practice (*Ellis, 2001*). Finally, the models only included anthropometric parameters; that is they did not consider other factors associated with body fat (*Moliner-Urdiales et al., 2009*; *Van Sluijs et al., 2010*; *Gómez-Martínez et al., 2012*). Nevertheless, even without these other factors our predictions were still precise.

## Comparison with the existing literature

When comparing our predictive models with the existing literature, we find studies that evaluate a single anthropometric parameter and other authors who developed multivariable predictive models based on skinfolds (*Silva et al., 2013*; *Jensen, Camargo & Bergamaschi, 2016*). We must bear in mind that the studies evaluating a single anthropometric parameter (circumferences of the upper arm, waist, hip and others; ratios such as waist-to-height or waist-to-hip) (*Freedman et al., 2004*; *Freedman et al., 2005*; *Freedman et al., 2012*; *Bergman et al., 2011*; *Goossens et al., 2012*; *Boeke et al., 2013*; *Weber et al., 2013*; *Craig et al., 2014*) did not assess the power of this parameter, the interactions with other variables, or a combination of factors to estimate body fat, which gives greater accuracy to the results obtained in our model. Multivariable models of using skinfolds have the clinical drawback of skinfolds being difficult to measure and the methodological drawback that they did not follow the statistical techniques that yield the highest power in the construction and validation of a multiple linear regression predictive model (interactions, powers, bootstrapping, ICC and testing of approximately 6,500 models) (*Silva et al., 2013*; *Jensen, Camargo & Bergamaschi, 2016*). In light of the above, our model clearly provides greater accuracy with respect to the others published in the scientific literature.

## Implications for research and practice

The diagnosis, treatment and follow-up of obesity in the pediatric age group have become a global health priority. Consequently, there is great interest in the healthcare field in the development of quick and accurate tools that can be used in the follow-up of these patients. In this study, we developed formulas for the calculation of body fat percentage and FMI based on BMI, age and sex, which facilitate monitoring of adiposity in the management of these patients, reserving the use of more accurate methods such as DXA for extreme cases.

Regarding possible future lines of research, we encourage other authors to externally validate the equations developed in this paper. The methodology used in this work can be applied to create new equations for body fat or for other types of parameters, both anthropometric and non-anthropometric.

## CONCLUSIONS

Body fat percentage and FMI measured by DXA can be accurately estimated in children and adolescents with overweight and obesity using our predictive models based on BMI, age and sex. Our models enable quick calculation of body fat percentage and FMI, thereby simplifying and reducing the use of resources in everyday clinical practice. We also highlight our methodology, which could be applied to obtain similar equations for the analyzed parameters.

## ACKNOWLEDGEMENTS

The authors thank Maria Repice and Ian Johnstone for their review of the English version of this paper.

### Funding

The authors received no funding for this work.

### Competing Interests

Antonio Palazón-Bru is an Academic Editor for PeerJ.

### Author Contributions

- Ernesto Cortés-Castell conceived and designed the experiments, wrote the paper, reviewed drafts of the paper.
- Mercedes Juste conceived and designed the experiments, performed the experiments, contributed reagents/materials/analysis tools, wrote the paper, reviewed drafts of the paper.
- Antonio Palazón-Bru conceived and designed the experiments, analyzed the data, wrote the paper, prepared figures and/or tables, reviewed drafts of the paper.
- Laura Monge conceived and designed the experiments, contributed reagents/materials/analysis tools, reviewed drafts of the paper.
- Francisco Sánchez-Ferrer conceived and designed the experiments, contributed reagents/materials/analysis tools, reviewed drafts of the paper.
- María Mercedes Rizo-Baeza conceived and designed the experiments, reviewed drafts of the paper.

### Human Ethics

The following information was supplied relating to ethical approvals (i.e., approving body and any reference numbers):

This study adhered to the ethical principles of the Declaration of Helsinki, and only involved the performance of procedures used in everyday clinical practice. The data were processed safeguarding anonymity and confidentiality, and the study was approved by the corresponding Ethics Committee (Comité Ético de Investigación Clínica del Hospital Universitario de San Juan de Alicante, ref 16/305). Informed consent was not requested from the parents for this study, since it was part of routine clinical practice without any type of intervention. The Ethics Committee approved this procedure.

### Data Availability

The data set has been uploaded as a Supplementary File.

## Supplemental Information

Supplemental information for this article can be found online at http://dx.doi.org/10.7717/peerj.3238#supplemental-information.

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
