# Peer review of "A simple equation to estimate body fat percentage in children with overweightness or obesity: a retrospective study"

_PeerJ, doi:10.7717/peerj.3238_

## Round 0.1 · original submission · Major Revisions

The three reviewers have raised concerns about the methodological aspects of the study, in particular the lack of generalizability due to the use of a convenience sample, and the evaluation of the prediction model.

Please ensure you address all of the reviewers' comments in the revised article, and include a point by point description of the changes that have been made.

Reviewer 1 ·

Basic reporting

In general this paper is well written and reported.

Experimental design

As indicated below I have some comments.

Validity of the findings

The findings may very well be valid but some additions are needed to confirm this.

Additional comments

Important points

1. Lines 63-64: I do not agree that DXA is considered as the gold standard method for the assessment of body fat. Instead a four-component model (Sopher A, Shen E, Peitrobeilli A. 2005. Pediatric body composition methods. In: Heymsfield S, Lohman T, Wang Z, editors. Human body composition, 2nd ed. Champaign, IL: Human Kinetics. p 129–140.) is generally considered to provide the most valid estimates of body composition. The problem with DXA is that it may be associated with bias. (Sopher AB, Thornton JC, Wang J, Pierson RN, Jr., Heymsfield SB, Horlick M. 2004. Measurement of percentage of body fat in 411 children and adolescents: a comparison of dual-energy X-ray absorptiometry with a four compartment
model. Pediatrics 113:1285–1290.; Williams JE, Wells JC, Wilson CM, Haroun D, Lucas A, Fewtrell MS. 2006. Evaluation of Lunar Prodigy dual-energy X-ray absorptiometry for assessing body composition in healthy persons and patients by comparison with the criterion 4-component model. Am J Clin Nutr 83:1047–1054.; Demerath EW, Fields DA. Am J Human Biology 26: 291-304, 2014.) This problem should be noted and discussed in the paper.

2. Table 1: The investigated subjects are apparently very fat with 43 % body fat on the average. Therefore the proposed equations are probably limited to overweight/obese children. Additional information should be given regarding the overweight/obesity status of the children used to establish these equations. Such information is needed for different age groups and sex. Without such information it is not possible to know in which cases the proposed equations could be applied.

3. I think the evaluation procedure used is interesting but nevertheless I think that also an evaluation based on the Bland & Altman procedure should be included (Bland JM, Altman DG. Statistical-Methods for Assessing Agreement between 2 Methods of Clinical Measurement. Lancet.1986; 1(8476): 307-10).This procedure is commonly used to evaluate body composition methods and provides important information.

Reviewer 2 ·

Basic reporting

Although the results of the study are relevant for practical application, there are some important concerns such as: inclusiveness of ages, lack of sensitivity/specificity measures, concerns over validations of predicted models, and limited applicability to norms not clinical populations.

Experimental design

The manuscript is generally well written and the discussion of previous publications appropriate. The different fat distribution between different races should also be mentioned as any equations developed may be racially specific. There are a small number of spelling errors.

Validity of the findings

I think you should perform a sensitivity and specificity analysis to see how well the regression equations predict body fat percentage. A correlation looks for similarities and you need also to look for differences.
The validation of the proposed prediction models should be evaluated by other methods , such as Bland Altman Plots or Clark error grid analysis. The model is only valid for normal growing children, whereas in daily clinic children with chronic diseases , presenting with aberrant anthropometric measures,are seen.
The same technician positioned the subjects, performed the three scans, and executed the analysis according to the operator’s manual using the standard analysis protocol.

Additional comments

If equations are proposed for this population, it should be noted whether subjects were excluded under conditions of overweight or obesity. Therefore, it is suggested that before generating the equations it should be noted whether these are generated for normal population or if subjects with malnutrition and obesity are included. It should be clarified.

Reviewer 3 ·

Basic reporting

This article is dealing with an interesting topic, that it, try to produce read-to-use formula in order to facilitate the work of clinicians and epidemiologists.
The manuscript is well written and the English language is correct.

Even if there are many important references throughout the manuscript, some important contributions related with body composition and children and adolescents are missing. I only cite some few.

Obes Sci Pract. 2016 Sep;2(3):272-281. Epub 2016 Jul 20.
Anthropometric predictors of body fat in a large population of 9-year-old school-aged children.
Almeida SM1, Furtado JM1, Mascarenhas P2, Ferraz ME3, Silva LR1, Ferreira JC4, Monteiro M5, Vilanova M6, Ferraz FP1.

Child Obes. 2016 Aug;12(4):314-23. doi: 10.1089/chi.2015.0020. Epub 2016 Apr 5.
Comparison of Eight Equations That Predict Percent Body Fat Using Skinfolds in American Youth.
Truesdale KP1, Roberts A1, Cai J2, Berge JM3, Stevens J1,4.
These authors conclude that predictive models should not be used in paediatrics.


In the introduction and discussion section, the debate is missing regarding the determinants of body fat. In this sense, some multicenter studies on representative samples should be cited, like the HELENA study in adolescents, the IDEFIX study in children, the EYHS (all European studies) ,the AVENA study in Spain, to cite also only a few. Socioeconomic factors, education of the mother, physical activity, physical fitness,among others, have been identified as determinants of body fat in children and adolescents.

Somo references should also be included, like the international standardization for BMI of Tim Cole and colleagues that was adopted by the IOTF.

Experimental design

The research could fit into the aims and scope of the journal, but not with the current approach.
The research question is clearly stated.
There could be a gap for the addressed research.

The sample is a convenience sample of children attending a hospital. This reviewer has great concerns about this aspect, as the procedure has not been done on a representative sample. Additionally, these children were refered to the hospital, so they are not representative at all.

The statistical approach is correct.

Validity of the findings

"The fat mass percentage formula had a coefficient of determination of 0.65. This value was
0.863 for FMI. In the validation, the constructed formulas had an intraclass correlation
coefficient of 0.77 for fat mass percentage and 0.92 for FMI."

The replicability of the proposed formula seems to low to me.

In fact, i have been trying some prediction on the provided excel sheet, and with the same height (1,68) and weight (55) and age (17), for a male body fat is 15% and a female 33%.

As Fat Free Mass is not included, the data could be equivocal.

Additional comments

The proposed formula can not be generalized for the general paediatric population. May be the authors could work on it in order it applies to children attending the hospital. Important determinants like hours of physical activity, fitness, should be taken into account.

I would recommend to use the word "sex" instead of "gender" as it is the biological aspect which is considered.

---

## Round 0.2 · Minor Revisions

The authors have satisfactorily addressed most of the reviewers' concerns. However, the following two issues need to be fixed:

Page 8, line 206: “Finally, the Bland & Altman procedure was satisfactory (Fig. 3).” - The Bland-Altman limits of agreement analysis was performed as requested by two reviewers but the figure needs the limits of agreement (+/- 1.96 SD) added (horizontal lines) and a proper interpretation in the Results and Discussion. Stating that the procedure “was satisfactory” is firstly not correct (we are interested in the _results_ not the _procedure_) and extremely vague.

Page 9, lines 222-224: “We also underscore the statistical methodology followed for its construction, which took into account the subject-to-variable ratio of about 6500 models.” - I must have missed this sentence in the original version. What does it mean? Where do the 6500 come from? This number re-appears later in the Discussion, again with no indication where it comes from.
Also, this and the following sentence are not strengths but method descriptions (“Bootstrapping was applied (reference method to validate predictive models) and the intraclass correlation coefficient was determined instead of Pearson's, since the estimate and the value obtained by DXA referred to the same variable (body fat or FMI) (Steyerberg et al., 2001).” Reword to make it clear why it is a strength or remove it.

There were also a couple of typos and odd wording choices that I corrected in the manuscript using tracked changes (see attachment). I also added a qualifier in a number of instances that the results apply only to children with overweight or obesity.

---

## Round 0.3 · accepted · Accept

All outstanding issues have been addressed by the authors.